# The effect of religiosity on violence: Results from a Brazilian population-based representative survey of 4,607 individuals

Juliane Piasseschi de Bernardin Gonçalves[1]*, Clarice Sandi Madruga[2], Giancarlo Lucchetti[3], Maria do Rosário Dias Latorre[4], Ronaldo Laranjeira[2], Homero Vallada[1]

1 Department of Psychiatry, Faculdade de Medicina da Universidade de Sao Paulo (LIM-23/ProSER), Sao Paulo, Brazil, 2 Psychiatry Department, Federal University of Sao Paulo, Sao Paulo, Brazil, 3 Department of Medicine, School of Medicine, Federal University of Juiz de Fora, Juiz de Fora, Brazil, 4 Department of Epidemiology, School of Public Health, University of Sao Paulo, Sao Paulo, Brazil

* juliane.pbg@usp.br

**Data Availability Statement:** All relevant data are within the paper and its Supporting Information files.

## Abstract

Although there is a wide array of evidence showing the beneficial effect of religiosity on violence among adolescents, nationwide studies in the general population are scarce. This study aims to explore whether religiosity is associated or not with diminishing violence in a Brazilian population-based representative sample. This observational cross-sectional study was conducted in 2011–2012 using face-to-face interviews and included 4,608 individuals 14 years and older. The survey included measures of religiosity (religious affiliation and importance of religion), violence (involvement in fights, domestic violence and police detention), depression, social support and alcohol dependence. We used logistic regression models and mediation analyses. In the total sample analyses, after adjustments, having a religious affiliation was inversely associated with lower involvement in fights (OR = 0.60, CI95%:0.37–0.98) and less police detention (OR = 0.37,CI95%:0.20–0.70), whereas the importance of religion was only associated with less fights (OR = 0.60,CI95%:0.36–0.99). Subanalyses revealed different associations depending on the age group evaluated. Mediation tests showed that the association of religious affiliation on violence outcomes was mediated by alcohol use. In conclusion, religiosity seems to be an important factor associated with lower levels of violence in this nationwide representative survey and alcohol dependence seems to mediate this relationship. Health professionals should be aware of these findings in their clinical practice.

## Introduction

The World Health Organization estimates that more than 1.3 million deaths occur every year as the result of violence, which is the fourth leading cause of death worldwide for individuals aged 15–44 years old [1]. A worldwide survey of 187 countries showed a 45% increase in firearm attacks and 32.6% increase in the use of sharp objects from 1990 to 2013 [2]. In addition,

**Funding:** This work received financial support from CNPq-Brazilian Council for Scientific and Technological Development (GL, HV and RL) and from CAPES-Brazilian Federal Government Agency (JPBG).

**Competing interests:** The authors have declared that no competing interests exist.

violence has a long-term impact on psychological and social outcomes, with economic consequences that include the high costs of treatment, the increased use of mental health services, emergency care and criminal justice [1].

Due to the high prevalence rates of violence, a growing number of preventive initiatives have been developed and implemented [3], which is supported by the latest version of the "Diagnostic and Statistical Manual of Mental Disorders" (DSM-V– 2013) of the American Psychiatric Association that suggests the inclusion of cultural perspectives to the treatment of victims of violence [4]. Among the complementary strategies proposed for mental health rehabilitation programs, there is evidence that interventions based on religiosity/spirituality (R/S) can be beneficial [5]. Likewise, studies are showing the promising role of R/S in diminishing and/or preventing violent behavior [6–8].

Several theories have been developed to explain the aforementioned relationship. The impact of R/S on violence has been noted in the 60s by Hirschi and Stark [9], who hypothesized the "Social control theory", in which religion could prevent individual criminal behavior through supernatural boundaries and rewarding Divine promises. This theory suggests that criminality can be prevented by socialization bonds and by the fear of derogatory acts creating a self-imposition behavior against violence.

Other authors have developed the "Costly signaling theory of religion", which proposes that "costly rituals" maybe "hard-to-fake" signals, incorporating an idea of a "cultural commitment" among a group of people. Since it is not possible to measure the degree of commitment, rituals that involve extensive time and effort may indicate that only individuals who are honestly devoted to the community will be able to fit in the group (in that case the religious one) [10,11]. Still attending the idea of an evolutionary point of view, another relevant hypothesis is the "Moral development theory", which is based on the improvement of consciousness regarding moral and spiritual beliefs as age passes by [12,13]. According to this theory, with aging, an individual goes through different stages of moral development in life, including changes and reflections about moral and spiritual issues, which can prevent from perpetrating violence as older as an individual gets.

Although further attempts in the 80s failed to confirm the relevancy of those hypothesis on diminishing violence [14,15], more recently, representative samples of American adolescents and young adults showed an inverse relationship between levels of R/S and delinquency [16–20].

Moreover, three systematic reviews including rigorous methodological designs showed that R/S may be inversely associated with several risk behavior outcomes in adolescents [6–8]. Despite this evidence, most authors consider involvement with drugs as a "delinquency behavior," since the use of those substances is prohibited. Therefore, Kelly et al. [8] performed a meta-analysis separating drug delinquency from nondrug delinquency, showing that the magnitude of R/S was different among the measures, as it was stronger for drug use, followed by alcohol use and then by delinquency. The separation into two groups (drug and nondrug delinquency) is a potential approach to further understand this relationship, since R/S has been constantly described as protective for substance use and abuse [5,21].

Despite the growing evidence of this relationship, mediators were also not fully explored by previous authors. Heavy episodic drinking was recently described as a partial mediator of church attendance on sexual aggression perpetration in a sample of male adolescents [22]. However, other drugs and social support, were not investigated as potential mediators of this relationship.

Within this context, this study aims to advance in this field of knowledge, investigating the general population and exploring possible mediators, which were seldom studied in other samples. In 2000, Benda and Toombs [23] reported a moderator effect of age, where

individuals with high levels of R/S above 24 years old were less inclined to perpetrate violence. Therefore, the present study aimed to investigate the relationship between R/S and involvement in fights, police detention and domestic violence in a Brazilian large nationally representative sample of general population.

## Methods

### Study design

This study was part of the Second Brazilian National Alcohol and Drugs Survey (II BNADS), a cross-sectional multistage cluster sampling of alcohol consumption patterns [24]. The protocol was approved by the Ethics Committee of the Federal University of Sao Paulo and by the Brazilian National Commission of Ethics in Research (CONEP), Brazil. All subjects provided written informed consent prior to being interviewed and, in the case of minors, consent from parents or guardians were obtained. All respondents granted their informed consent and the interviews complied with all statements required by the Brazilian Ministry of Health Ethical Committee Office (CAAE: 61909615.0.0000.5505) and the Ethics Committee of the Federal University of Sao Paulo.

### Setting, participants and eligibility criteria

This Brazilian representative population-based survey was conducted between November 2011 to March 2012 during a face-to-face interview in the respondents' homes with individuals who were at least 14 years old. The choice for including those with less than 18 years old was based on a previous publication that found high levels of substance and alcohol use among pre-adolescents in Brazil [25], even though our laws prohibit the sale of alcoholic beverages to minors.

The process involved three steps: 1) selection of 149 counties using probability proportional to size methods (PPS); 2) selection of 375 census sectors (two for each county), also using PPS; and 3) selection of eight simple random households within each census sector, followed by sample selection of a household member by the "closest future birthday" technique. Individuals with severe cognitive deficiency, those who were absent after three visits and who did not agree to participate as well as those from native Brazilian tribes were not included. The reasons for not include indigenous populations can be justified by the fact that these individuals live in far regions with very limited access, there are ethical and legal restrictions for the inclusion of this vulnerable population in research in Brazil, native indigenous groups represent less than 1% of our total population and at least 18% of them do not speak Portuguese language [26].

### Interview

The one-hour interview was carried out by trained research assistants and was an adaptation of the "Hispanic Americans Baseline Alcohol Survey (HABLAS)" [27], fully checked for its consistency. In order to guarantee the quality of the data collection, 20% of the interviews were followed by the principal researchers.

### Variables used in this study

**Dependent variables.** Three different dependent variables of violence were analyzed:

- Involvement in fights: assessed through the question: "In the last twelve months, how often were you involved in a fight with physical aggression?" Answers were dichotomized as "never" (0) or "at least once" (1).

- Domestic violence: only asked for people married/living with a partner: "In the last twelve months, did you do any of these things to your partner: throw things, push, shake, slap, bite, kick, burn, force to have sex or strike with a knife/weapon?" Answers were dichotomized as "never" (0) or "at least once" (1). Domestic violence represented 56% of the total sample including people between 31–59 years of age.

- Police detention: assessed through the question "In the last twelve months, have you been detained or arrested by the police?" It could be answered by "yes" (1) or "no (0)."

   **Independent variables.**   Religiosity was assessed in the following way:

- Religious affiliation: appraised using the question "What is your religion?" with the most prevalent religions as possible answers, which were categorized as "none" (0) or "had any" (1).

- Importance of religion: evaluated using the question "How important is the religion in your life?" Answers were categorized as "not important/indifferent/without religion" (0) and "very important/important" (1).

   - Other measures were:

- Depression: through the validated Brazilian version of the 20-item Center for Epidemiological Studies Depression Scale (CES-D) [28]. The score ranged from 0 (no depression) to 60 and was classified as "yes" or "no" by the cutoff of 16 points.

- Social support: using an adaptation of the Adult Psychiatric Morbidity Survey of England and categorized results as "high" or "low" using 17 points as the cutoff [29].

- Alcohol dependence: was diagnosed using the DSM-V criteria and classified as "yes" or "no" according to the criteria of the CIDI cutoff [30].

- Sociodemographic variables: age (which was subdivided in four age group: 14–18, 19–30, 31–59, 60 and above), gender (female and male), years of school, Brazilian regions (North, Northeast, Midwest, Southeast, South), marital status (married/living with a partner, single/divorced/widower) and race (classified as: White, Black, Brown and others). Race in Brazil was assessed by the self-report of the color of skin with "Brown" referring to a mix of different races (predominantly refers to European and African ancestries but could also be related to Amerindians and African ancestries).

## Statistical methods

STATA 13.0 was used for all analyses which were weighted due to the stratified sampling design and non-responses using post-stratification. All individuals who refused or answered "I do not know" in any of the questions were excluded, and they represented 0.2% of the total sample. First, a descriptive analysis was carried out using absolute number and percentages, means and standard deviations for each dependent and independent variable.

   To analyze the relationship between violence and religiosity, we used chi-square tests and logistic regression models (OR-*odds ratio*) with 95% confidence intervals. Models were reported in an unadjusted and in an adjusted way for sociodemographic variables (age, gender, education, marital status), depression, social support and alcohol dependence. These models were verified in the general population and also in age group sub analysis.

Finally, in order to verify the possible mediator role of different independent variables in the relationship between religious variables and violence outcomes, we carried out a series of mediation models [31]. To explore this mediation models, Chi-square tests (CI:95%) were used to verify if the variables met the following criteria: a) religiosity was associated with mediators; b) religiosity was associated with violence without mediators; c) mediators had a significant effect on violence; and d) the effect of religiosity on violence was attenuated after addition of the mediator [32]. If the criteria were met, Sobel-Goodman tests [33] were used to identify if there was a partial or total mediation effect. Partial mediation is showed when there are significant levels on both direct and indirect pathways and total mediation means that there is only significance on indirect effect [34]. A p<0.05 was considered significant in all analyses.

## Results

The dataset was composed of 4,607 individuals, representing 77% of the total sample (response rate). Sociodemographic characteristics, violence and mental health outcomes are presented in Table 1. The proportions of men and women were similarly distributed. The mean age of the total sample was 35.8 years (SD = 18.8), and the average period of formal education was 8.8 (SD = 4.7) years. Most people (57.2%) lived with a partner and the predominant races were Brown (44.3%) and White (40.1%).

Domestic violence was the most prevalent category of aggressive behavior (8.4%) in the past twelve months, followed by involvement in fights (2.6%) and arrests by the police (1.4%). The total sample population showed high levels of social support (71%). Regarding mental health, depression was the most prevalent mental health condition (25%), and alcohol dependence were diagnosed in fewer than 10% of the responders.

More than 90% of the participants had a religious affiliation and 80.9% considered it "very important to their lives" (Table 2). In the bivariate analysis (Chi-square tests), having a religious affiliation was significantly associated with less violent outcomes, whereas the importance attributed to religion was only associated with less fights.

The regression models between violence variables and religiosity are presented in Table 3. After all adjustment, religious affiliation showed an inverse association with fights (*OR* = 0.60; 95%CI [0.37–0.98]) and police detention (*OR* = 0.37; 95%CI [0.20–0.70]), but not with domestic violence (*OR* = 0.57; 95% CI [0.30–1.08]). On the other hand, the importance of religion was also significantly associated with less involvement in fights (*OR* = 0.60; 95% CI [0.36–0.99]).

Table 4 presents age group models. Domestic violence could not be stratified in the age groups category as it was already stratified regarding marital status (only the participants who were married/living with a partner). Religious affiliation among adolescents [14–18] was inversely associated with violent behaviors in all analyses performed, losing significance when alcohol dependence was added. In the age group ranging from 19 to 30 years old, religious affiliation was inversely associated with police detention even after all adjustments. Finally, in the age group between 31 to 59 years old, only the importance of religion was associated with less involvement in fights, finding not maintained after adjustments.

In the mediation analyses, the incorporation of social support variable in the regression model produced no significant changes in the effect of religiosity; social support did not have a mediating role in this model. On the other hand, stratified analyses pointed that alcohol dependence could have a mediation role between religiosity and violence outcomes. Therefore, to explore the mediation effect, we used chi-square (95%CI) test to determine the association between alcohol dependence and religious affiliation ($\chi^2$ = 80.57; p = 0.001), importance of religion ($\chi^2$ = 67.26; p = 0.001), involvement in fights ($\chi^2$ = 135.11; p = 0.001), police detention

**Table 1. Characteristics of sociodemographic, violence and mental health variables.**

| Variable | Total Sample N = 4607 | % Weighted |
|---|---|---|
| **Sociodemographic** | | |
| *Sex* | | |
| Male | 2,070 | 47.9 |
| Female | 2,537 | 52.1 |
| *Age* | | |
| 14–18 | 1,229 | 11.9 |
| 19–30 | 952 | 26.0 |
| 31–59 | 1,779 | 47.3 |
| 60–99 | 647 | 14.8 |
| *Marital Status* | | |
| Single/separated/widowed | 1,907 | 31.8 |
| Living with partner | 2,120 | 57.18 |
| *Race* | | |
| White | 1,828 | 40.1 |
| Black | 570 | 12.6 |
| Brown | 2,074 | 44.3 |
| Others | 126 | 3.0 |
| *Brazilian regions* | | |
| North | 417 | 7.4 |
| Northeast | 1,262 | 27.2 |
| Midwest | 317 | 7.2 |
| Southeast | 1,926 | 43.2 |
| South | 685 | 14.9 |
| **Violence** | | |
| *Involvement in Fights* | | |
| No | 4,442 | 97.4 |
| Yes | 160 | 2.6 |
| *Police Detention* | | |
| No | 4,536 | 98.6 |
| Yes | 66 | 1.4 |
| *Domestic Violence* | | |
| No | 1,926 | 91.6 |
| Yes | 191 | 8.4 |
| **Mental Health** | | |
| *Social Support* | | |
| Low | 1,357 | 28.9 |
| High | 3,250 | 71.1 |
| *Depression* | | |
| Yes | 1,167 | 25.1 |
| No | 3,338 | 74.9 |
| *Alcohol dependence* | | |
| Yes | 390 | 10.0 |
| No | 4,217 | 90.0 |

($\chi^2$ = 53.95; p = 0.001) and domestic violence ($\chi^2$ = 107.22; p = 0.001). This variable met all criteria for inclusion as a possible mediator.

Fig 1 shows the analyses of the mediation effect tested for religious affiliation with involvement in fights, police detention and domestic violence, and the mediation effect of the

**Table 2. Description of the religiosity on different outcomes of violence.**

| | | | Involvement in fights | | | | | Police detention | | | | | Domestic violence[1] | | | | |
|---|---|---|---|---|---|---|---|---|---|---|---|---|---|---|---|---|---|
| | | | Yes | | No | | | Yes | | No | | | Yes | | No | | |
| Variables | Total Sample N = 4607 | %* | N | %* | N | %* | p-value ‡ | N | %* | N | %* | p-value ‡ | N | %* | N | %* | p-value ‡ |
| *Religious Affiliation* | | | | | | | | | | | | | | | | | |
| Catholic | 2,888 | 64 | 72 | 1.2 | 2,813 | 62.8 | **0.001** | 40 | 0.8 | 2,846 | 63.2 | **0.001** | 119 | 5.3 | 1,251 | 59.9 | **0.001** |
| Protestant | 1,108 | 23.5 | 41 | 0.7 | 1,065 | 22.8 | | 8 | 0.1 | 1,099 | 23.4 | | 40 | 1.6 | 482 | 22.3 | |
| Others | 153 | 3.1 | 10 | 0.2 | 143 | 8.7 | | 2 | 0 | 150 | 3.1 | | 1 | 0 | 64 | 0.3 | |
| None | 448 | 9.3 | 36 | 0.6 | 412 | 3 | | 16 | 0.4 | 431 | 8.9 | | 29 | 1.3 | 127 | 6.1 | |
| *Importance of Religion* | | | | | | | | | | | | | | | | | |
| Very important | 3,369 | 80.9 | 80 | 1.4 | 3,285 | 79.5 | **0.001** | 42 | 1 | 3,324 | 80 | 0.826 | 127 | 6 | 1,503 | 77.2 | 0.183 |
| Important | 579 | 14.2 | 27 | 0.4 | 551 | 13.7 | | 7 | 0.1 | 572 | 14 | | 24 | 1.1 | 221 | 11.3 | |
| Indifferent | 175 | 4.1 | 11 | 0.3 | 164 | 3.8 | | 1 | 0 | 173 | 4.1 | | 9 | 0.4 | 66 | 3.2 | |
| Few/not important | 36 | 0.7 | 6 | 0 | 30 | 0.6 | | 0 | 0 | 36 | 0.7 | | 2 | 0.1 | 9 | 0.5 | |

* = weighted percentage.

‡ = p-value performed by Chi-square test (CI:95%).

[1] = data from married people or living with partner.

importance of religion with involvement in fights. Religious affiliation was totally mediated by alcohol dependence in relation to its effect on domestic violence. However religious affiliation was partially mediate by alcohol dependence in involvement in fights and police detention and the importance of religion was partially mediated by alcohol dependence in involvement in fights.

**Table 3. Associations between different outcomes of violence and religiosity.**

| | Outcome | | | | | |
|---|---|---|---|---|---|---|
| | *Involvement in fights* | | *Police detention* | | *Domestic violence*[1] | |
| | *No* | *Yes* | *No* | *Yes* | *No* | *Yes* |
| *Religious Affiliation* | | | | | | |
| *Crude* | 1.00 | 0.33 (0.20–0.55)* | 1.00 | 0.26 (0.13–0.52)* | 1.00 | 0.33 (0.21–0.67)* |
| *OR (a)* | | 0.46 (0.28–0.77)** | | 0.31 (0.15–0.63)** | | 0.48 (0.25–0.89)* |
| *OR (b)* | | 0.49 (0.30–0.79)** | | 0.30 (0.15–0.61)** | | 0.46 (0.25–0.88)* |
| *OR (c)* | | 0.49 (0.30–0.79)** | | 0.30 (0.15–0.61)** | | 0.47 (0.25–0.89)* |
| *OR (d)* | | 0.60 (0.37–0.98)* | | 0.37 (0.20–0.70)** | | 0.57 (0.30–1.08) |
| *Importance of Religion* | | | | | | |
| *Crude* | 1.00 | 0.34 (0.20–0.56)* | 1.00 | DN/A | 1.00 | DN/A |
| *OR (a)* | | 0.45 (0.26–0.74)** | | | | |
| *OR (b)* | | 0.46 (0.28–0.75)** | | | | |
| *OR (c)* | | 0.48 (0.29–0.78)** | | | | |
| *OR (d)* | | 0.60 (0.36–0.99)* | | | | |

[1] Not adjusted for marital status.

a) Adjusted for sociodemographic variables (age, gender, education, marital status, family income). b) Adjusted for sociodemographic and mental health variables (age, gender, education, marital status, family income, depression). c) Adjusted for sociodemographic and mental health variables and social support. d) Adjusted for sociodemographic and mental health variables (including social support) and alcohol dependency.

*: $p < 0.05$.

**: $p < 0.01$.

DN/A: do not apply.

**Table 4. Associations between violence outcomes and religiosity stratified by age group, Brazil, 2012.**

| | Outcome | | | |
| --- | --- | --- | --- | --- |
| | *Involvement in fights* | | *Police Detention* | |
| | **No** | **Yes** | **No** | **Yes** |
| *Religious Affiliation* | | | | |
| 14–18 (a) | 1.00 | 0.42 (0.20–0.86)* | 1.00 | 0.24 (0.08–0.69)** |
| 14–18 (b) | | 0.61 (0.30–1.23) | | 0.42 (0.12–1.53) |
| 19–30 (a) | | 0.57 (0.21–1.50) | | 0.20 (0.06–0.63)* |
| 19–30 (b) | | 0.65 (0.24–1.72) | | 0.22 (0.07–0.65)** |
| 31–59 (a) | | 0.52 (0.17–1.62) | | 1.40 (0.18–10.58) |
| 31–59 (b) | | 0.66 (0.19–2.31) | | 2.00 (0.28–14.43) |
| *Importance of Religion* | | | | |
| 14–18 (a) | 1.00 | 0.48 (0.27–0.85)* | | DN/A |
| 14–18 (b) | | 0.59 (0.33–1.04) | | DN/A |
| 19–30 (a) | | 0.57 (0.24–1.36) | | DN/A |
| 19–30 (b) | | 0.73 (0.29–1.83) | | DN/A |
| 31–59 (a) | | 0.39 (0.18–0.86)* | | DN/A |
| 31–59 (b) | | 0.49 (0.22–1.13) | | DN/A |

a) Adjusted for sociodemographic variables (age, gender, education, marital status, depression, support social). b) Adjusted for (a) and alcohol dependence.

*: $p < 0.05$.

**: $p < 0.01$.

DN/A: do not apply.

## Discussion

In this nationwide representative population based-study of 4,607 Brazilian individuals from the general population, religiosity was inversely associated with different violent outcomes, in that individuals having a religious affiliation were less involved in fights and detention; and those who considered religion important in their lives were also less involved in fights. This relationship seems to be mediated by alcohol dependence and is potentially different among age groups. These results could add to the current scientific literature and will be explained further below.

Our findings corroborate with previous studies which showed that organizational religiosity (such as religious affiliations) may have a larger effect on violence than self-reported religiosity (such as the importance of religion in an individual's life) [8]. The first explanation for this effect is the role that religion can play as hypothesized in the "Social control theory". Basically, a formal religion usually engages individuals in volunteering and philanthropic behaviors, as well as provides psychosocial adjustment and social support, most commonly presenting non-violent behaviors as its doctrine or thinking's [35,36], which could minimize violent behaviors.

Providing support to the play of social control hypothesis, Resnick et al. [18] investigated fights, threats, stabbings and shootings in a representative sample of American adolescents and concluded that youths who attended religious services and had greater value for their religion were less involved in violence. In the same line, a previous study found that not having a religion was a risk factor for intimate partner violence in Brazil [37] and others found that carrying a weapon was inhibited among individuals with high levels of R/S [23,38]. Religious

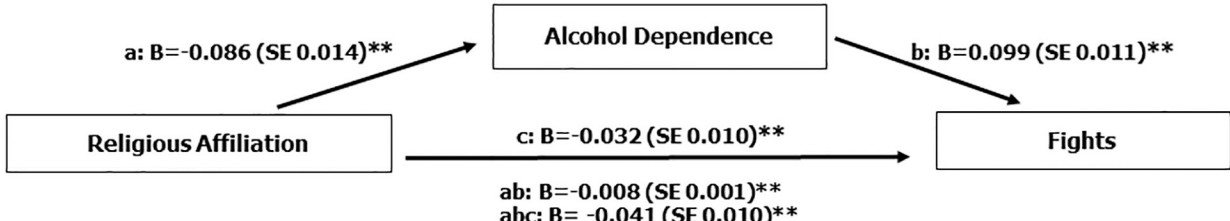

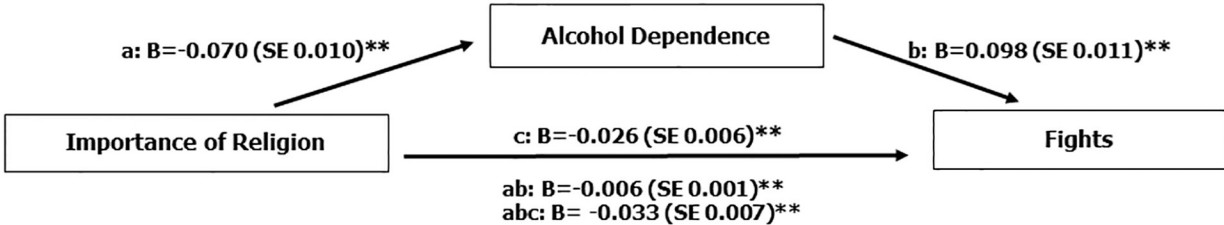

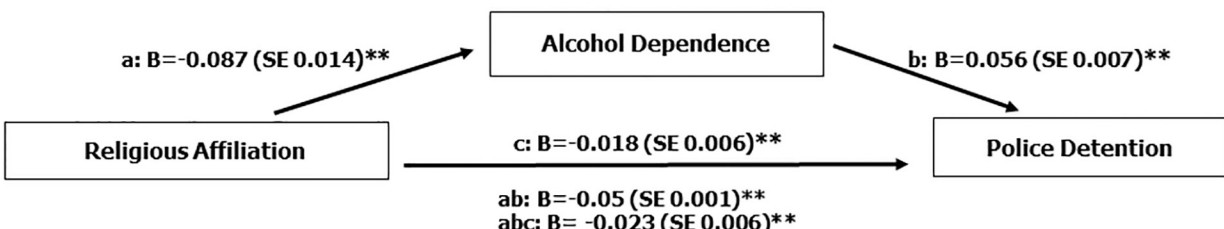

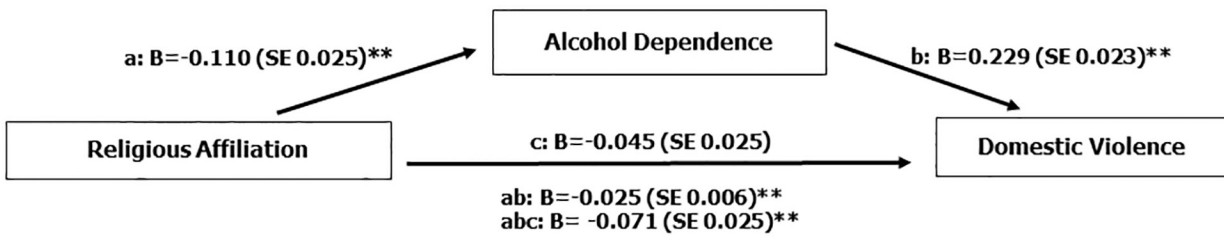

**Fig 1. Alcohol dependence as a mediator: Indirect effect of religiosity on violence.**

social control theory seems to support those findings independently of the violence behavior perpetrated.

A second explanation for these findings is available at the "Cost signaling theory of religion", which stablishes that only those who are truly committed and involved on sharing ideas

of a specific group, will be able to renounce pleasant or productive activities in order to be part of the group, and for this reason, will have a more generously behavior into public life [10,39].

A third explanation is based on the moral essay´s "human development theory", which suggests that as age advances to a more reflective stage of life, more prominent the essence of religion, meaning and purpose of life becomes [12,40]. The impact of the importance of religion was more noted in adults than in adolescents. These results are supported by previous studies which found less engagement in gangs and fights for inmates above 24 years of age who had private religiosity, but not for those who attended religious services [23]. Another study noted that a decrease in theft was predominantly the result of religious attendance in adolescents and the result of private religiosity in young adults [41]. The attribution of importance to religion is considered a private religiosity, a concept independent of a formal religion, which can raise consciousness to prevent destructive social behavior [42].

In addition, R/S is a multidimensional concept consisting of different pathways to influence human behavior. Concerning the role of mediators in this relationship, social development factors, such as antisocial bonding and beliefs, were entirely responsible for mediating different measures of R/S and violence on a sample of adolescents [38]. To our knowledge, however, no study has explored social support as an indirect effect of religiosity on violence. It is possible that encouraging positive social behavior may be a more distinguished pathway in which religiosity may operate.

Alcohol dependence, on the other hand, had an important role as mediator, attenuating religiosity and maintaining the effect in different age groups. In a study using the first database of BNADS demonstrated that individuals with high levels of R/S had more supporting opinions concerning alcohol control policies, such as limiting hours of sales and advertisements, raising the legal drinking age and increasing taxes [21]. It is possible that the prohibitive role of religion on alcohol as a licit drug is more evident than that in illicit drugs, which can be more of a secular institutional problem to be controlled.

The observed association between religious affiliation on domestic violence in entirely mediated by the effect of alcohol abuse/dependence. These findings corroborated by previous findings, which showed that heavy episodes of drinking mediated the protective effect of church attendance on sexual aggression in a follow-up period of 2 years in university students [22] and by a study that showed that discussing spiritual/moral values to partners, victims and perpetrators of domestic violence, resulted in a 71% reduction in the cases of violence [43]

It is not unusual to find modest results of R/S mediation analysis [44,45], probably because of the complexity of this multidimensional concept. We suggest further investigation of other mental health disorders as possible mediators of R/S and violence, such as antisocial and borderline personality disorders, which can be associated with more aggressive behaviors [46]. Furthermore, interventional methodological designs to reduce aggressive and criminal behavior have been showing satisfactory results [43,47] and should be considered when exploring possible mechanisms of action of the R/S dimension.

## Limitations

The present study has some limitations that should be considered. First, although we assessed two important and previously studied variables [7], it is notable that the R/S dimension should be treated as a multidimensional concept and assessed in different measures. The use of more detail and validated questionnaires to assess R/S instead of the use of single questions could have found different results. Second, our study design was cross-sectional and, therefore, no inferences of causal-effect relationship could be made.

Third, although this is a nation-wide representative survey, it reflects the Brazilian context. Some authors suggest that cultural aspects may influence the religiosity and violence relationship [35,43]. Although the most recent meta-analysis did not find differences between studies performed in the United States of America and studies in other countries [8], it is worth mentioning that Brazil is a highly religious country [48]. Thus, cultural influences should be considered when interpreting the results of the present study.

Forth, although previous studies have supported our decision on asking about domestic violence only to individuals who were living with a partner [37], it is possible that those having a relationship, but not living together could have perpetrated violence against the partner as well. Likewise, this variable might be less significant to a young population, since it requires the individuals to be married or living with a partner.

Finally, the analyses were conducted using dichotomous variables. This approach was performed in order to facilitate and to give power to our analyses, since the percentages of these events were relatively low. However, our results could be flattened (i.e. those who were perpetrators of violence could be doing so at a higher rate than what was allowed in the variables).

## Conclusions

Religiosity seems to be an important factor associated with lower levels of violence in this nationwide representative survey, across all age groups investigated. Alcohol dependence seems to mediate this relationship. Health professionals should be aware of these findings in their clinical practice. The role of the religious/spiritual dimension in violence needs to be further investigated in order to justify its inclusion of complementary medical approaches.

## Supporting information

**S1 Data.**
(XLSX)

## Acknowledgments

We would like to thank Cleusa Ferri for her comments and advice on this work.

## Author Contributions

**Conceptualization:** Juliane Piasseschi de Bernardin Gonçalves, Ronaldo Laranjeira, Homero Vallada.

**Formal analysis:** Juliane Piasseschi de Bernardin Gonçalves, Giancarlo Lucchetti, Maria do Rosário Dias Latorre.

**Funding acquisition:** Ronaldo Laranjeira, Homero Vallada.

**Methodology:** Clarice Sandi Madruga, Ronaldo Laranjeira, Homero Vallada.

**Project administration:** Clarice Sandi Madruga, Ronaldo Laranjeira, Homero Vallada.

**Writing – original draft:** Juliane Piasseschi de Bernardin Gonçalves.

**Writing – review & editing:** Clarice Sandi Madruga, Giancarlo Lucchetti, Maria do Rosário Dias Latorre, Ronaldo Laranjeira, Homero Vallada.

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
