## [Decision Letter · Decision Letter 0]

27 May 2020

PONE-D-20-08721

The effect of religiosity on violence: results from a nationwide population-based representative survey of 4,607 individuals.

PLOS ONE

Dear Dr. Gonçalves,

Thank you for submitting your manuscript to PLOS ONE. After careful consideration, we feel that it has merit but does not fully meet PLOS ONE’s publication criteria as it currently stands. Therefore, we invite you to submit a revised version of the manuscript that addresses the points raised during the review process.

We look forward to receiving your revised manuscript.

Kind regards,

Laura Schwab-Reese

Academic Editor

PLOS ONE

Reviewers' comments:

Reviewer's Responses to Questions

**Comments to the Author**

1. Is the manuscript technically sound, and do the data support the conclusions?

Reviewer #1: Yes

Reviewer #2: Yes

2. Has the statistical analysis been performed appropriately and rigorously? 

Reviewer #1: Yes

Reviewer #2: Yes

3. Have the authors made all data underlying the findings in their manuscript fully available?

Reviewer #1: No

Reviewer #2: Yes

4. Is the manuscript presented in an intelligible fashion and written in standard English?

Reviewer #1: Yes

Reviewer #2: Yes

5. Review Comments to the Author

Reviewer #1: This article contributes needed additional findings to the field of violence prevention and religion. I am excited to see this work is being conducted internationally, and I hope that the authors continue to conduct this research. Overall the manuscript is well written with only a few issues with proofreading. There are concerns I have though, related to lack of information provided on the theories mentioned in the discussion as well as some methodological concerns. Additionally, PLOS ONE requires that authors make their data underlying the findings fully available, and from the manuscript at this time, this did not appear to be present. Below are my comments/concerns related to the specifics within the manuscript.

Dichotomous Scales: Limitations Need to be Addressed

o By doing them as dichotomous the results could be flattened. Those who were perpetrators of violence could be doing so at a higher rate than what was allowed in the variables. This is a limitation of the study.

Variables: Limitations Need to be Addressed

• Variables asked Domestic Violence only to those who live together: This is a limitation as people are in relationships outside of living together. Therefore, more individual could had been included if this question was asked to any individuals who were in a partnership.

Sampling: Limitations Need to be Addressed

. Large national studies tend to not include their indigenous populations which can have a negative impact (i.e. lack of awareness of issues they experience, assistance). I know that the Kayapo are a indigenous population native to Brazil. It would be good to explain why the Kayapo and other indigenous people were not sampled.

o I am assuming that Brazil has different consent rules for their IRB than other countries, so the inclusion of those under 18 should be expanded upon. How and why they are included is important to indicate especially since the variables being examined might be less significant to a young population such as the domestic violence questions- requiring the individuals to be married or living with a partner.

By this variable being restricted it is possible that those who are younger and potentially perpetrators of DV on their partners were not included in this analysis.

Discussion Concerns

o You did not measure religious attendance and only measured if they indicated they were religious and it was/was not important. Unsure if this sentence (Page 16; Lines 240-242) can be supported with your measures and findings.

o It might be good to look into the literature on costly signaling and religion. This is an area of research in biological anthropology and provides context to this sentence (Page 15; Lines 244-246). At this point, this sentence seems to oversimplify why religion can function as a means to control/limit violent behaviors. Adding literature on costly signaling will help support your argument.

o Religious social control theory (Page 16; Line 252). This needs to be introduced as an theory earlier on in the introduction. There is no explanation as to what religious social control theory is, and the reader is expected to know what this theory is from the previous paragraph. There need to be more build up and support as to what specifically this theory is and then use the information above as support.

o The explanation of this theory (Human development theory) is confusing and requires more clarity (Page 16; Line 254-257). How dose human development theory connect to religion? What is the theory first- then move into connection to your findings. This theory as the one above should be referenced in some way during the literature review.

Reviewer #2: Spelling errors:

Line 36- Were used

line 121 where

lines 125,129, 137, ?". remove the period after the quote marks

line 296 uso

Other than the spelling/grammatical errors, I found it to be clear, concise, statistically sound, and a nice addition to the previous research.

6. PLOS authors have the option to publish the peer review history of their article (what does this mean?). If published, this will include your full peer review and any attached files.

Reviewer #1: No

Reviewer #2: No

---

## [Author Response · Author response to Decision Letter 0]

3 Aug 2020

Response to Reviewers:

ANSWER: We have checked the manuscript and it is complying with PLOS ONE's style requirements.

ANSWER: We have now included this information in the manuscript: “All subjects provided written informed consent prior to being interviewed and, in the case of minors, consent from parents or guardians were obtained”. See page 5 and 6, lines 113 – 115.

ANSWERS TO ITEMS 3a AND 3b: We have followed the protocols of different research ethics committees concerning data availability: the Federal University of São Paulo (UNIFESP), Brazilian National Ethics Research Committee (CONEP) and the São Paulo State Drug Policy Committee (CONED), which follows all recommendations regarding the use of the database for partial studies. Therefore, the survey dataset use requests are analyzed by the Brazilian National Alcohol and Drugs Survey (BNADS) technical committee. Data requests may be sent to the survey’s coordinator clarice.madruga@unifesp.br or via our website: www.inpad.org.br/lenad.

Reviewers' comments:

Reviewer's Responses to Questions

Review Comments to the Author

Reviewer #1: This article contributes needed additional findings to the field of violence prevention and religion. I am excited to see this work is being conducted internationally, and I hope that the authors continue to conduct this research. Overall the manuscript is well written with only a few issues with proofreading. There are concerns I have though, related to lack of information provided on the theories mentioned in the discussion as well as some methodological concerns. Additionally, PLOS ONE requires that authors make their data underlying the findings fully available, and from the manuscript at this time, this did not appear to be present. Below are my comments/concerns related to the specifics within the manuscript.

ANSWER: Thank you very much for your comments and suggestions in order to improve our manuscript. You will found our responses to your concerns below. Proofreading was also carried out as requested. In relation to the data availability, unfortunately, our Ethics Research Committee does not allow online data sharing due to ethical reasons, since we have included minors (participants with less than 18 years old). Nevertheless, according to the guidelines of Plos One, we have now referred an institutional body to which data requests may be sent. 

Dichotomous Scales: Limitations Need to be Addressed

o By doing them as dichotomous the results could be flattened. Those who were perpetrators of violence could be doing so at a higher rate than what was allowed in the variables. This is a limitation of the study.

ANSWER: We agree with your comment and we have now included this limitation as suggested. See page 16, lines 336 – 340.

Variables: Limitations Need to be Addressed

• Variables asked Domestic Violence only to those who live together: This is a limitation as people are in relationships outside of living together. Therefore, more individual could had been included if this question was asked to any individuals who were in a partnership.

ANSWER: We have now included this potential limitation as well. See page 15, lines 331 - 335.

Sampling: Limitations Need to be Addressed

. Large national studies tend to not include their indigenous populations which can have a negative impact (i.e. lack of awareness of issues they experience, assistance). I know that the Kayapo are a indigenous population native to Brazil. It would be good to explain why the Kayapo and other indigenous people were not sampled.

ANSWER: We have now explained why indigenous groups were not included in this large representative study. See page 6, lines 133 – 137.

o I am assuming that Brazil has different consent rules for their IRB than other countries, so the inclusion of those under 18 should be expanded upon. How and why they are included is important to indicate especially since the variables being examined might be less significant to a young population such as the domestic violence questions- requiring the individuals to be married or living with a partner.

ANSWER: We agree with you that more information is needed concerning the inclusion of minors in the sample. We have now included the consent rules for this sample (consent from guardians or parents were obtained) and we have justified why this group of participants were included. See pages 5 and 6, lines 113 – 115. 

By this variable being restricted it is possible that those who are younger and potentially perpetrators of DV on their partners were not included in this analysis.

ANSWER: We agree with your comment and decided to include this information as a limitation of the study. See page 6, lines 123 – 126.

Discussion Concerns

o You did not measure religious attendance and only measured if they indicated they were religious and it was/was not important. Unsure if this sentence (Page 16; Lines 240-242) can be supported with your measures and findings.

ANSWER: We agree that the inclusion of “religious attendance” could be misleading to the reader in this sentence. We have modified it accordingly, including only “religious affiliations” in the organizational religiosity. See page 12, lines 259 – 261. 

o It might be good to look into the literature on costly signaling and religion. This is an area of research in biological anthropology and provides context to this sentence (Page 15; Lines 244-246). At this point, this sentence seems to oversimplify why religion can function as a means to control/limit violent behaviors. Adding literature on costly signaling will help support your argument.

ANSWER: Thank you for your important reference. We have now added the costly signing theory as suggested. See page 4, lines 71 – 81; page 13, lines 275 – 279.

o Religious social control theory (Page 16; Line 252). This needs to be introduced as an theory earlier on in the introduction. There is no explanation as to what religious social control theory is, and the reader is expected to know what this theory is from the previous paragraph. There need to be more build up and support as to what specifically this theory is and then use the information above as support.

ANSWER: We have now included the theories upfront in the Introduction. This will allow the reader to get a closer contact with our hypotheses right in the beginning of the manuscript. See page 3, lines 65 – 70; page 12, lines 262.

o The explanation of this theory (Human development theory) is confusing and requires more clarity (Page 16; Line 254-257). How dose human development theory connect to religion? What is the theory first- then move into connection to your findings. This theory as the one above should be referenced in some way during the literature review.

ANSWER: We have now clarified the “Human development theory”, introducing it to the reader in the Introduction and further explaining it in the Discussion. See page ; page 13, lines 280 – 282.

Grammar corrections:

Page 3, Line 54 – Is this really overuse or increased use?

ANSWER: We have modified this sentence. Page 3, line 55 – overuse – changed by increased use.

Page 18, line 296 – use?

ANSWER: We have modified this sentence. Page 15, line 322 – uso – changed for use.

Page 18, line 299 – Brazilian context

ANSWER: We have modified this sentence. Page 15, line 326 – scenario changed for context.

Page 18, line 300 – influence the

ANSWER: We have modified this sentence. Page 15, line 326 – interfere changed for “influence the”.

Reviewer #2: 

Spelling errors:

Page 2, Line 36- Were used

ANSWER: We have modified this sentence. See page 2, line 37 – were changed to we.

Page 6, line 121 where

ANSWER: We have corrected it as suggested. See page 5, line 149 – where changed to were.

Page 6, lines 125,129, 137, ?". remove the period after the quote marks

ANSWER: Periods were removed as requested. See page 5, lines 149, 153, 157.

Page 16, line 296 – uso

ANSWER: We have now corrected it. Page 15, line 322 – uso – changed for use.

Other than the spelling/grammatical errors, I found it to be clear, concise, statistically sound, and a nice addition to the previous research.

ANSWER: Thank you very much for your comments and suggestions.

---

## [Editor Report · Decision Letter 1]

10 Aug 2020

The effect of religiosity on violence: results from a Brazilian population-based representative survey of 4,607 individuals.

PONE-D-20-08721R1

Dear Dr. Gonçalves,

We’re pleased to inform you that your manuscript has been judged scientifically suitable for publication and will be formally accepted for publication once it meets all outstanding technical requirements.

Kind regards,

Laura Schwab-Reese

Academic Editor

PLOS ONE
---

## [Editor Report · Acceptance letter]

13 Aug 2020

PONE-D-20-08721R1 

The effect of religiosity on violence: results from a Brazilian population-based representative survey of 4,607 individuals. 

Dear Dr. Gonçalves:

I'm pleased to inform you that your manuscript has been deemed suitable for publication in PLOS ONE. Congratulations! Your manuscript is now with our production department. 

Kind regards, 

on behalf of

Dr. Laura Schwab-Reese 

Academic Editor

PLOS ONE